# Portevin-Le Chatelier Characterization of Quenched Al-Mg Alloy Sheet with Different Mg Concentrations

**DOI:** 10.3390/ma15144965

**Published:** 2022-07-17

**Authors:** Ni Tian, Wenze Wang, Zhen Feng, Weihao Song, Tianshi Wang, Zijie Zeng, Gang Zhao, Gaowu Qin

**Affiliations:** 1School of Materials Science & Engineering, Northeastern University, Wenhua Road, No. 3-11, Heping District, Shenyang 110819, China; 2070509@stu.neu.edu.cn (W.W.); 2070348@stu.neu.edu.cn (Z.F.); 2100571@stu.neu.edu.cn (W.S.); 20192958@stu.neu.edu.cn (T.W.); 20193231@stu.neu.edu.cn (Z.Z.); zhaog@mail.neu.edu.cn (G.Z.); qingw@smm.neu.edu.cn (G.Q.); 2Key Laboratory for Anisotropy and Texture of Materials (Ministry of Education), Northeastern University, No. 3-11, Wenhua Road, Heping District, Shenyang 110819, China

**Keywords:** Al-Mg alloy sheet, quenched, Mg content, Portevin-Le Chatelier effect, room-temperature tensile

## Abstract

In the present study, the PLC characteristic parameters and DSA mechanism of Al-(2.86~9.41) Mg alloy sheets were investigated during tensile testing at room temperature with a tensile rate of 1 × 10^−3^ s^−1^. On the basis of the solution Mg concentrations in the α-Al matrix, the initial vacancy concentration, the second-phase particle configuration and the recrystallized grain configuration are almost the same by quenching treatment. The results show that the type of room-temperature tensile stress–strain curves of quenched Al-(2.86~9.41) Mg alloy sheets varied according to the Mg content. The type of stress–strain curve of the Al-2.86 Mg alloy sheet was B + C, while the type of stress–strain curve of the Al-(4.23~9.41) Mg alloy sheets was C. When the quenched Al-(2.86~9.41) Mg alloy sheets were stretched at room temperature, the strain cycle of the rectangular waves corresponding to the high stress flow $\Delta \varepsilon_{T}^{\max}$ and stress drop amplitude Δσ on the zigzag stress–strain curve of alloy sheets increased with increasing the Mg content. Moreover, the strain cycle of $\Delta \varepsilon_{T}^{\max}$ and Δσ on the stress–strain curve of alloy sheets increased gradually with increasing tensile deformation. The yield stress of quenched Al-(2.86~9.41) Mg alloy sheets increased gradually with increasing the Mg content. Moreover, the critical strain corresponding to yield stress $\varepsilon_{\sigma }$ and the critical strain corresponding to the occurrence of the PLC shearing band $\varepsilon_{c}$ of alloy sheets both increased with increasing the Mg content. However, the difference in flow strain value $\Delta \varepsilon_{c-\sigma }$ between $\varepsilon_{c}$ and $\varepsilon_{\sigma }$ of alloy sheets decreased gradually with increasing the Mg content.

## 1. Introduction

The transient interaction between solute atoms and dislocation motion during tensile deformation of Al-Mg alloys at room temperature, called Dynamic Strain Aging (DSA for short), leads to a serrated oscillation phenomenon with increasing flow strain at a certain strain rate and temperature, called the Portevin-Le Chatelier (PLC) effect. [1,2,3]. In recent decades, many studies have investigated the influence of solid solution atomic diffusion [4,5] and tensile rate [2,6] on the PLC characteristic parameters of aluminum alloys. Peng et al. [7] found that there was a critical temperature Tt = 49 °C for the tensile deformation of an annealed 3034 aluminum alloy between −50~200 °C. When the tensile temperature was below 49 °C, the critical strain corresponding to the occurrence of the PLC shearing band $\varepsilon_{c}$ was negatively correlated with the tensile temperature because the DSA was mainly controlled by the interaction between solute Mg atoms and movable dislocations; that is, the $\varepsilon_{c}$ decreased gradually with increasing tensile temperature. When the tensile temperature was higher than 49 °C, the critical strain $\varepsilon_{c}$ was positively correlated with the tensile temperature because the concentration of Mg atoms in the matrix was reduced due to the segregation of Mg atoms or the formation of Mg-containing alloy phases; that is, the $\varepsilon_{c}$ increased gradually with increasing tensile temperature. A. Chatterjee et al. [8] found that the tensile strain rate increased from 8.02 × 10^−5^ s^−1^ to 2.02 × 10^−3^ s^−1^, the PLC effect characteristic parameters of Al-2.5 Mg alloy, such as the average stress drop amplitude and average waiting time, decreased while the strain cycle corresponding to the stress oscillation increased, and the serrated type of tensile flow curve changed from type B to type A.

Previous findings [9] showed that the PLC effect characteristic parameters of 5083 aluminum alloy sheets show obvious differences with the variation in the forms of Mg atoms in the matrix. There was an obvious yield platform in the flow curves when the annealed Al-(2.86~9.41) Mg alloy sheets were stretched at a tensile rate from 1 × 10^−4^ s^−1^ to 1 × 10^−2^ s^−1^ at room temperature, and the strain range corresponding to the yield platform increased with increasing strain rate. However, there was no yield platform in the tensile curves of the as-quenched Al-(2.86~9.41) Mg alloy sheets. On the other hand, when the tensile strain rate was lower than 1 × 10^−4^ s^−1^, the waiting time for the PLC effect to appear on the tensile curve of the annealed alloy sheet was longer than that of the quenched alloy sheet, but the critical strain for the PLC effect to appear on the tensile curve of the annealed sheet was lower than that of the quenched sheet. The essential reason for the PLC effect in aluminum alloys is the interaction between solute atoms and dislocation motion. Due to the existence of Mg atoms in the matrix, the number and distribution of Mg atoms have a significant effect on the mechanical behavior of the aluminum alloy sheet and the PLC characteristic parameters, which determine the mechanical properties of the alloy sheet [10,11]. Wen et al. [12] found that the precipitation of the β phase resulted in the desolation of solution Mg atoms in a 5182 aluminum alloy matrix when the annealing time was extended from 30 h to 100 h at 182 °C. Therefore, the stress drop amplitude of the PLC effect on the tensile flow curve of the alloy at room temperature decreased. Picu et al. [13,14] put forward a new DSA theory on the basis of an atomistic analysis of Mg diffusion along the core of dislocations in Al. Moreover, it was believed that the main cause of the PLC effect in Al-Mg alloy is the large number of vacancies generated by jog motion, which quickened Mg atom diffusion velocity, promoted solute Mg atom segregation on the moving dislocations and then pinning the dislocations effectively. Solute atom diffusion along the dislocation pipe was as difficult as that in the matrix when the vacancies in the alloy were in a very dilute concentration. However, the intertwining of the movable dislocation and the dislocation forest formed jog, and the large number of vacancies generated by jog motion and the solute atom segregation on the forest dislocation hindered dislocation motion and then resulted in an increase in flow stress. As the flow stress increases, the movable dislocation can escape the pinning of solute atom segregation and continue to move, which results in a decrease in flow stress. However, as the dislocation continues to slip, it would encounter the other solute atoms and become pinned again, leading to another increase in stress. The above process cycle starts over and repeats itself, forming the PLC effect. Sarkar et al. [15] found that the excess vacancy concentration in the Al-2.5 Mg alloy sheet increased obviously after being soluted at 400 °C for 16 h and then water quenched, which led to an increase in the number of stress drop occurrences per unit time on the tensile flow curve at room temperature. That is, the strain cycle corresponding to the stress oscillation of the PLC effect was significantly reduced as the vacancy concentration in the alloy matrix increased. K. Chihab et al. [16] found that both the critical strain rate for the variation in PLC curves from type B to type A and the critical strain for the occurrence of the PLC effect in the tensile curve increased with increasing Mg content from 0.9% to 2% when the Al-(0.9~2.0) Mg alloy sheets were stretched at room temperature. However, the PLC phenomenon, the PLC characteristic parameters such as the strain cycle corresponding to the stress oscillation, stress drop amplitude, critical strain corresponding to the occurrence of the PLC shearing band and the interaction mechanism between solute Mg atoms and moving dislocations in high-magnesium aluminum alloy sheets are not well understood.

In the present study, the Al-(2.86~9.41) Mg aluminum alloy sheets were subjected to solution and then water quenching to control the solute Mg concentrations in the α-Al matrix, but the initial vacancy concentration, the second-phase particle configuration and the recrystallized grain configuration were approximately the same; therefore, the influence of the solute Mg atom concentration on the characteristic parameters of PLC of the Al-Mg alloy sheet was investigated by stretching at room temperature with a tensile rate of 1 × 10^−3^ s^−1^, to reveal the mechanism of DSA in a high-magnesium aluminum alloy sheet.

## 2. Materials and Methods

The commercial purity aluminum and commercial purity magnesium were smelted in a graphite crucible resistance furnace in this study, and ingots with a size of 32 mm × 18 mm × 12 mm were cast in a water-cooled copper mold; the casting temperature was 720 °C ± 3 °C. The specific alloy composition is shown in Table 1. The ingots were homogenized at 540 °C for 24 h in a circulation air resistance furnace and were hot rolled at 480 °C into 1.5 mm-thick alloy sheets. The tensile specimens were cut from the alloy sheets along the rolling direction, and the specific size of the tensile specimens is shown in Figure 1. All the tensile specimens had a solid solution treatment at 450 °C for 2 h in a circulation air resistance furnace and then were allowed to quench in water. Then, a tensile test at room temperature (25 °C) was carried out immediately afterward.

The tensile experiments were conducted using a Shimadzu AG-X100 electronic universal testing machine (Shimadzu Corporation, Kyoto, Japan) at room temperature (25 °C) for strain rates of 1 × 10^−3^ s^−1^, and all samples were stretched within 20 min. The yield strength and tensile strength of the specimens were measured with an extensometer with a gauge length of 50 mm, and then the sawtooth number of the flow curve in the strain range (Δε = 0.003) was determined from the true stress-true strain curve. The type of PLC curves could be determined by calculating the strain cycle of the flow stress oscillation. The characteristic parameters, including the strain cycle corresponding to the PLC effect of the tensile flow curve, the strain cycle of the rectangular waves corresponding to the high stress flow $\Delta \varepsilon_{T}^{\max}$, the strain cycle of triangle waves corresponding to the low stress flow $\Delta \varepsilon_{T}^{\min}$, the stress drop amplitude Δσ, the critical strain corresponding to the occurrence of the PLC shearing band $\varepsilon_{c}$ and the difference between the critical strain variable of PLC occurrence and the critical strain corresponding to the yield of the alloy sheet $\Delta \varepsilon_{c-\sigma }$ = ($\varepsilon_{c}-\varepsilon_{\sigma }$), were measured. Therefore, the PLC effect of the alloy could be quantitatively analyzed.

The content of each element in the four alloys were used by direct reading spectrometer (FMP; FOUNDRY-MASTER PRO C0213, Oxford Instruments, Oxford, UK), and the final results were the average of three tests, and the equipment has an error of 0.03–0.08 in the detection of Mg content, 0.007 in the detection of Fe content, 0.004 in the detection of Si content and 0.001 in the detection of Zn content. The morphology, phase distribution and grain structure of the samples were observed by optical microscopy (OM; OLYMPUS GX71, Olympus Corporation, Tokyo, Japan) and focused ion beam scanning electron microscopy (SEM, ULTRA PLUS, Carl Zeiss AG, Oberkochen, Germany). Quantitative analysis of atomic concentration and qualitative analysis of phase composition in aluminum alloy matrix was carried out by Energy Disperse Spectroscopy (EDS; X-MAX^N^(50), Oxford Instruments, Oxford, UK), and the detector has an experimental error of 0.1%. The grain structure observation samples were anodic coated with film at a voltage of 25 V, and the composition of the coating solution was 1% HF + 1% HBF_4_ + 24% C_2_H_5_OH + 74% H_2_O.

## 3. Results

### 3.1. Effect of Mg Content on the Microstructure of Quenched Al-(2.86~9.41) Mg Alloy Sheets

Figure 2 shows the optical microstructure of quenched Al-(2.86~9.41) Mg alloy sheets. It can be seen that a large number of micron-sized gray second-phase particles, which were the Al Mg Fe crystal phase, formed in the solidification and crystallization process of the alloy ingot [4]. After hot rolling, the particles were broken and distributed in a streamline in the aluminum matrix. No significant change was found in the morphology, quantity, and distribution of the micron-sized Al Mg Fe phase particles with increasing Mg content.

Figure 3 shows the SEM images and the results of the energy spectrum analysis of quenched Al-(2.86~9.41) Mg alloy sheets. The second-phase particles of the gray–black polygonal second-phase particles of 0.25–2.5 μm in the alloy sheets were the Al Mg Fe phase [4], which is an insoluble alloy phase precipitated by crystallization during the casting solidification process. It cannot be dissolved into the matrix by homogenization treatment, and the subsequent hot rolling will break it into broken chains. Meanwhile, Figure 3 shows that there were no β phase particles in the quenched Al-(2.86~9.41) Mg alloy sheets, which was consistent with the research results of Zhao et al. [17]. This result indicated that Mg existed in the form of solute atoms in the quenched Al-(2.86~9.41) Mg alloy sheets. It is worth noting that the results of the energy spectrum analysis show that the mass fractions of Mg atoms in the matrix of alloy sheets with Mg contents of 2.86%, 4.23%, 6.51% and 9.41% were 2.82%, 4.19%, 6.48% and 9.37%, respectively. This was consistent with the results of Goswani et al. [5]. Figure 3e show the result of EDS analysis of Al Mg Fe phase in the Al-4.23Mg alloy sheet.

Figure 4 shows the OM images of grains of quenched Al-(2.86~9.41) Mg alloy sheets. The grains of the quenched sheets with different Mg contents were equiaxed grains with a size of approximately 25 μm. This result indicated that full recrystallization occurred in the Al-(2.86~9.41) Mg alloy sheets after holding at 450 °C for 2 h, which was consistent with the results of Nogueira et al. [18] and Gubiza et al. [19]. No obvious change was found in the shape and size of recrystallized grains with the increase in Mg content from 2.86% to 9.41%.

### 3.2. Effect of Mg Content on the Stress–Strain Curve Features of Aluminum Alloy Sheets

Figure 5 shows the stress–strain curves of the quenched alloy sheets with different Mg contents at a 1 × 10^−3^ s^−1^ strain rate at room temperature and the enlarged local images corresponding to different strain stages. Figure 5b–d shows enlarged local diagrams of the flow stress–flow strain curves corresponding to ε < 0.035 (the initial strain $\varepsilon_{\mathrm{Ti}}$), 0.035 < ε < 0.065 (the middle strain $\varepsilon_{\mathrm{Tm}}$) and ε > 0.065 (the late strain $\varepsilon_{\mathrm{Tl}}$), respectively. The tensile stress–strain curves of the alloy sheets with different Mg contents all showed obvious zigzag shapes, that is, the PLC phenomenon. Great discrepancies can be seen in the yield stress of the alloy sheet and the serration streamline characteristics of the stress–strain curves with different Mg contents. When the Mg content increased from 2.86% to 9.41%, the flow stress corresponding to the same strain increased gradually, and the yield strength of the alloy sheets increased from 85 MPa to 139 MPa. The flow stress of the alloy sheets with Mg contents of 2.86%, 4.23% and 6.51% after the yield point with increasing flow strain, namely, an oscillation phenomenon of “increase-fall-increase”, repeatedly increased, and the amplitude of the flow stress oscillation increased gradually first and then basically remained unchanged with increasing strain. It is worth noting that the stress–strain curve of Al-9.41 Mg alloy sheet at the yield stage had zigzag features. When the strain exceeded 0.0147, the stress–strain curve was smooth, and the stress increased continuously with increasing strain; however, when the strain exceeded 0.142, the flow stress of Al-9.41 Mg alloy sheet showed an oscillation phenomenon of “increase-fall-increase” repeatedly with increasing flow strain, and the amplitude of the flow stress “increase-fall-increase” oscillation first increased gradually and then basically remained unchanged with increasing strain.

In addition, the rule of strain cycle $\varepsilon_{T}$ corresponding to the flow stress oscillation varied from the increase in strain when the quenched Al-(2.86~9.41) Mg alloy sheet was stretched at room temperature. The strain cycle $\varepsilon_{T}$ of the oscillation of the tensile flow stress of Al-2.86 Mg sheet showed a trend of decreasing first and then increasing slightly with increasing flow strain. The strain cycle $\varepsilon_{T}$ of the oscillation of the tensile flow stress of the Al-(4.23~9.41) Mg sheet showed a trend of decreasing gradually with increasing flow strain. Moreover, the strain cycle $\varepsilon_{T}$ of the oscillation of the tensile flow stress of alloy sheets was different at different tensile strain stages. When 0.002 < ε < 0.035, the Al-2.86 Mg, Al-4.23 Mg and Al-6.51 Mg alloy sheets were tensile, and the strain cycles of the flow stress oscillation were 2.7 × 10^−4^, 2.1 × 10^−4^ and 2.3 × 10^−4^, respectively. When 0.035 < ε < 0.065, the Al-2.86 Mg, Al-4.23 Mg and Al-6.51 Mg alloy sheets were tensile, and the strain cycle $\varepsilon_{T}$ of the flow stress oscillation was 2 × 10^−4^, 2.3 × 10^−4^ and 3 × 10^−4^, respectively. When 0.065, Al-2.86 Mg, Al-4.23 Mg and Al-6.51 Mg alloy sheets were tensile, the strain cycle $\varepsilon_{T}$ of flow stress oscillation was 2.5 × 10^−4^, 3.3 × 10^−4^ and 5 × 10^−4^, respectively. It is worth noting that for the Al-9.41 Mg sheet, the tensile curve corresponded to a strain between 0.002 < ε < 0.015, and the strain cycle $\varepsilon_{T}$ of tensile flow stress oscillation was 3 × 10^−4^. When 0.0147 < ε < 0.142, the stress–strain curve of the alloy sheet was smooth, namely, there was no PLC phenomenon. When ε > 0.104, the strain period $\varepsilon_{T}$ of the peak flow stress oscillation increased to 6 × 10^−4^. Because the strain period $\varepsilon_{T}$ of the peak flow stress oscillation of the Al-2.86 Mg alloy sheet first decreased and then increased slightly with increasing stress. Based on the results obtained by Zhou et al. [20], it was determined that the PLC line form of the Al-2.86 Mg alloy sheet was type (B + C). However, the strain period $\varepsilon_{T}$ of the tensile stress oscillating with increasing tensile strain of the Al-(4.23~9.41) Mg alloy showed a tendency to increase gradually. Therefore, the PLC line form of the Al-(4.23~9.41) Mg alloy sheet was type C.

Figure 5e–g shows local magnification diagrams at the initial, middle and late stages of tensile deformation about flow stress–strain of four Mg content alloy sheets. The curve waveform of the Al-(2.86~9.41) Mg alloy sheet at room temperature tensile testing, in which the flow stress oscillates with increasing strain, was asymmetric. The lower part of the waveform was triangular, while the upper part of the waveform was rectangular, which meant that the tensile flow stress of the alloy sheet gradually dropped to the lowest value and then gradually increased to the peak value in the process of tensile deformation, but the flow stress was maintained for a period of strain during the high stress stage. That is, in a certain strain stage, the moving dislocation was pinned continuously. Furthermore, the strain period of triangle wave $\Delta \varepsilon_{T}^{\min}$ (start-stop range of strain from the peak flow stress falling to rising again to the peak flow stress) and the continuous strain period of rectangular wave $\Delta \varepsilon_{T}^{\max}$ (start-stop range of strain when the flow stress is maintained at peak stage) were different with the increase of flow strain and Mg content. As a result, when the alloy sheets with different Mg contents were stretched at room temperature, the degree to which moving dislocations were pinned continuously was different in the process of tensile deformation, and the amount of deformation caused by the sliding of moving dislocations carrying a large number of Mg atoms was also different. At the same time, the corresponding strain range of the flow stress starting to drop and then climbing to the peak stress was also different when the PLC phenomenon occurred in the alloy sheets. That is, the strain variable caused by the rapid sliding when the motion dislocation broke the Mg atoms binding at room temperature stretching of the alloy sheets with different Mg contents and the number of sliding dislocations on the alloy surface when Mg atoms gradually converged near the moving dislocation and Mg atoms slip together with the moving dislocation until the flow stress reached the peak again were both different, so the strain variables of alloy sheets were different. This was similar to the experimental results of Picu et al. [13], Rizzi et al. [21] and Jiang et al. [22]. Figure 5h shows that the strain period of the oscillating rectangular wave $\Delta \varepsilon_{T}^{\max}$ corresponding to different strain stages increased first and then basically remained unchanged with increasing strain when the Al-(2.86~9.41) Mg alloy sheet stretched at room temperature.

The PLC phenomenon occurred only in the range of 0.002 < ε < 0.015 and ε > 0.065 in the tensile deformation curve of the Al-9.41 Mg alloy sheet at room temperature. Therefore, the variation law of the strain cycle of the rectangular waves corresponding to the high stress flow $\Delta \varepsilon_{T}^{\max}$ on the PLC curve of the quenched Al-(2.86~9.41) Mg alloy sheet with increasing Mg content was studied by dividing the tensile curve of the alloy sheet into four strain stages: 0.002 < ε < 0.015, 0.015 < ε < 0.035, 0.035 < ε < 0.065 and ε > 0.065. The results are shown in Figure 6. It can be seen that the strain cycle of rectangular waves corresponding to high stress flow $\Delta \varepsilon_{T}^{\max}$ on the PLC curve of alloy sheet increased gradually with the increase of Mg content at the same strain stage, and the larger the strain variable was, the greater the magnitude of the gradual increase of the strain period that the flow stress is pinned continuously with the increase of the Mg content. When 0.002 < ε < 0.015, the strain period was shorter when the flow stress of the Al-2.86 Mg alloy was maintained at the peak stage, which was only 0.9 × 10^−4^. When the Mg content increased to 4.23% and 6.51%, the strain period corresponding to the flow stress maintained at the peak stage increased slightly, and the $\Delta \varepsilon_{T}^{\max}$ gradually increased to 1.1 × 10^−4^. As high as 9.41%, the strain cycle of the rectangular waves corresponding to the high stress flow $\Delta \varepsilon_{T}^{\max}$ increased to 1.4 × 10^−4^. When 0.015 < ε < 0.035, the strain cycle of the rectangular waves corresponding to the high stress flow $\Delta \varepsilon_{T}^{\max}$ of the Al-2.86 Mg alloy sheet was only 1.2 × 10^−4^. With the Mg content increased to 4.23% and 6.51%, the strain cycle of rectangular waves corresponding to high stress flow $\Delta \varepsilon_{T}^{\max}$ increased to 1.3 × 10^−4^ and 1.4 × 10^−4^, respectively. When 0.035 < ε < 0.065, the strain period of the continuous high stress stage when the flow stress of the Al-2.86 Mg alloy sheet varied with the strain was 1.2 × 10^−4^. As the Mg content increased to 4.23% and 6.51%, the strain cycle of the rectangular waves corresponding to the high stress flow $\Delta \varepsilon_{T}^{\max}$ increased rapidly to 1.4 × 10^−4^ and 2 × 10^−4^, respectively. When ε > 0.065, the strain cycle of the rectangular waves corresponding to the high stress flow $\Delta \varepsilon_{T}^{\max}$ of the Al-2.86 Mg alloy sheet was 1.4 × 10^−4^. When the Mg content increased to 4.23% and 6.51%, the strain cycle of the rectangular waves corresponding to the high stress flow $\Delta \varepsilon_{T}^{\max}$ increased significantly to 1.9 × 10^−4^ and 2.8 × 10^−4^, respectively, and tended to be stable. When the Mg content increased to 9.41%, the strain cycle of the rectangular waves corresponding to the high stress flow $\Delta \varepsilon_{T}^{\max}$ maintained at the peak stage was the longest, reaching 3.1 × 10^−4^.

Figure 7 shows the change curve of the stress drop amplitude Δσ of the PLC phenomenon on the tensile curve of the quenched Al-(2.86~9.41) Mg alloy sheet at different strain stages as the Mg content increased. The change in the strain cycle of the rectangular waves corresponding to the high stress flow $\Delta \varepsilon_{T}^{\max}$ of the PLC curves of quenched Al-(2.86~9.41) Mg alloy sheet deformation was similar to the Mg content in Figure 6. When the degree of tensile deformation was the same, the stress drop amplitude Δσ of the tensile stress of the alloy sheet, which increased with increasing strain, increased with increasing the Mg content in the alloy sheet. With the increase in the strain variable, the amplitude of the stress oscillation Δσ increased gradually with increasing the Mg content. When 0.002 < ε < 0.015, the tensile flow stress drop amplitude Δσ of the Al-2.86 Mg alloy sheet was approximately 1.0 MPa. Under this condition, when the Mg content of the alloy sheet increased to 4.23% and 6.51%, its Δσ increased to 1.3 MPa and 1.4 MPa, respectively. When the Mg content increased to 9.41%, the stress drop amplitude Δσ was approximately 1.6 MPa. When 0.015 < ε < 0.035, the tensile flow stress drop amplitude Δσ of the Al-2.86 Mg alloy sheet was approximately 1.5 MPa, when the Mg content increased to 4.23% and 6.51%, the Δσ increased to 2.7 MPa and 2.8 MPa, respectively. When 0.035 < ε < 0.065, Δσ of the Al-2.86 Mg alloy sheet is 2.0 MPa, when the Mg content increased to 4.23% and 6.51%, Δσ increased to 3.3 MPa and 4.1 MPa, respectively. When ε > 0.065, the Δσ of the Al-2.86 Mg alloy sheet is 2.7 MPa, and when the Mg content increased to 4.23%, 6.51% and 9.41%, the tensile flow stress drop amplitude Δσ of the Al-2.86 Mg alloy sheet increased significantly to 4.1 MPa, 6.2 MPa and 7.7 MPa, respectively.

Figure 8 shows the change relationship curve of the critical strain corresponding to the occurrence of the PLC shearing band $\varepsilon_{c}$ and the critical strain corresponding to the yield stress $\varepsilon_{\sigma }$ of the Al-(2.86~9.41) Mg alloy sheet as the Mg content increased. The PLC phenomenon occurred only after the tensile yield plastic deformation of the quenched Al-(2.86~9.41) Mg alloy sheet. The critical strain corresponding to the occurrence of the PLC shearing band $\varepsilon_{c}$ and the critical strain corresponding to the yield stress $\varepsilon_{\sigma }$ of the alloy sheets increased with increasing Mg content. However, the difference in flow strain value $\Delta \varepsilon_{c-\sigma }$ between $\varepsilon_{c}$ and $\varepsilon_{\sigma }$ of alloy sheets decreased monotonically with increasing Mg content. The $\varepsilon_{c}$ and $\varepsilon_{\sigma }$ of the Al-2.86 Mg alloy sheets were 2.09 × 10^−3^ and 1.4 × 10^−3^, respectively. The difference flow strain value $\Delta \varepsilon_{c-\sigma }$ between $\varepsilon_{c}$ and $\varepsilon_{\sigma }$ of alloy sheets was 7.1 × 10^−4^. The $\varepsilon_{c}$ of the Al-4.23Mg alloy sheet increased to 2.2 × 10^−3^, and the $\varepsilon_{\sigma }$ increased to 1.6 × 10^−3^. The difference flow strain value $\Delta \varepsilon_{c-\sigma }$ between $\varepsilon_{c}$ and $\varepsilon_{\sigma }$ of alloy sheets was 6.3 × 10^−4^. When the Mg content increased to 6.51% and 9.41%, the $\varepsilon_{c}$ of the alloy sheet increased to 2.4 × 10^−3^ and 2.9 × 10^−3^, and the $\varepsilon_{\sigma }$ increased to 1.9 × 10^−3^ and 2.7 × 10^−3^, respectively. The difference flow strain value $\Delta \varepsilon_{c-\sigma }$ between $\varepsilon_{c}$ and $\varepsilon_{\sigma }$ of alloy sheets decreased to 5 × 10^−4^ and 4.7 × 10^−4^, respectively. The amount of strain that the alloy sheet bore from yielding to the PLC phenomenon began to decrease gradually with increasing the Mg content.

## 4. Discussion

No obvious differences were observed in the shape, size, quantity and distribution of the remaining constituent particles of the micron-sized Al Mg Fe or in the shape and size of grains in the Al-(2.86~9.41) Mg alloy sheets subjected to quenching treatment at 450 °C. However, the number of solution Mg atoms in the matrix of the alloy sheets increased as the Mg content increased from 2.86% to 9.41% (as shown in Figure 2, Figure 3 and Figure 4). The variation in the Mg content has no effect on the configuration of the micrometer remaining constituent Al Mg Fe in the as-quenched alloy sheets because the number of solution Mg atoms in the α-Al matrix of Al-(2.86~9.41) Mg was increased; however, the Fe content in the Al-(2.86~9.41) Mg alloy sheets was approximately the same. Meanwhile, Al-(2.86~9.41) Mg alloy sheets were recrystallized completely after solution treatment at 450 °C, which indicates that the increase in the concentration of solution Mg atoms in α-Al has no obvious effect on the recrystallization behavior of the α-Al matrix. Therefore, the grain configuration of the Al-(2.86~9.41) Mg alloy sheet was almost the same.

However, the solute Mg concentration in the α-Al matrix had a significant effect on the dislocation motion of the alloy sheet during tensile deformation at room temperature, which caused the difference in the type of tensile stress–strain curves of the as-quenched Al-(2.86~9.41) Mg alloy sheets. The stress–strain curve of the Al-2.86 Mg alloy sheet was type B + C, while those of the Al-(4.23~9.41) Mg alloy sheets were type C (as shown in Figure 5). The solute Mg atoms in the Al-2.86 Mg alloy sheet are in a very dilute concentration, which causes a small lattice distortion of the α-Al matrix and acts as weak obstacles to the dislocation motion. This is not only the main reason for the low strength but also for the wide strain range of the long strain cycle of the stress oscillation of the alloy sheet at the initial stage of tensile deformation. At the initial stage (0.002 < ε < 0.035) during tensile formation at room temperature, there is a small number of solution Mg atoms on the moving dislocation line in the Al-2.86 Mg alloy sheet, and the pinning force of the moving dislocation by the solution Mg atoms is weak. The maximum flow stress for the moving dislocation to break free from the pinning Mg atoms is low, and it is easy for the moving dislocation to break free from the pinning of solution Mg atoms and continue to move. Therefore, the strain cycle of the rectangular waves corresponding to the high stress flow $\Delta \varepsilon_{T}^{\max}$ of the alloy sheet was small. Meanwhile, the number of solution Mg atoms in the matrix of the Al-2.86 Mg alloy sheet was small, the distribution of solution Mg atoms in the α-Al matrix was sparse, and the depinning dislocation slipped off for a long distance before being again pinned by solution Mg atoms. Therefore, the strain cycle of triangular waves corresponding to the low stress flow $\Delta \varepsilon_{T}^{\min}$ of the alloy sheet was large. When the tensile strain was increased to the range of 0.035 < ε < 0.065, the vacancy concentration in the α-Al matrix of the alloy sheet increased. This can promote the diffusion of solution Mg atoms and gradually increase the number of solution Mg atoms segregating on dislocations. Then, the pinning force of solution Mg atoms on moving dislocations increased gradually, which caused the strain cycle of the rectangular waves corresponding to the high stress flow $\Delta \varepsilon_{T}^{\max}$ of the alloy sheet to increase slightly. It is easy for the moving dislocation to break free from the solution Mg atoms and continue to move; however, the moving dislocation will be pinning again soon for the rapid diffusion of solution Mg atoms, which causes the strain cycle of triangular waves corresponding to the low stress flow $\Delta \varepsilon_{T}^{\min}$ on the flow stress–strain curve of alloy sheet to decrease significantly. Under the comprehensive function of the two aspects mentioned above, the strain cycle $\varepsilon_{T}=\Delta \varepsilon_{T}^{\max}+\Delta \varepsilon_{T}^{\min}$ that the flow stress of the alloy sheet oscillates with the flow strain decreased obviously with increasing tensile strain. In other words, the serration oscillation frequency of the flow stress “climb-drop-climb” repeatedly changed from low to high gradually with increasing strain. Therefore, the stress–strain curve of Al-2.86 The Mg alloy sheet was type B + C.

The concentration of solution Mg atoms in the α-Al matrix increased with the Mg content of the alloy sheet by more than 4.23%, and the lattice distortion of the α-Al matrix and the pinning force of solid solution atoms on moving dislocations also increased. This is the main reason both for the increase in the strength of the alloy sheet and for the narrow strain range of the long strain cycle of the stress oscillation of the alloy sheet at the initial stage of tensile deformation (0.002 < ε < 0.035) as the Mg content increased. That is, the strain cycle ($\varepsilon_{T}$) of stress oscillation on the stress–strain curve of Al-(4.23–9.41) Mg alloy sheets is narrower than that of the Al-2.86 Mg alloy sheet because the number of solution Mg atoms segregated on the moving dislocation of alloy sheet increased with the increase in Mg content, the pinning force of moving dislocation by segregated Mg atoms also increased, and it is more difficult for the moving dislocation to break free from the pinning of solution Mg atoms. The moving dislocation would keep moving by carrying those solute Mg atoms as the flow stress was increased; therefore, the strain cycle of the rectangular waves corresponding to the high stress flow $\Delta \varepsilon_{T}^{\max}$ on the flow stress curve of the alloy sheet increased. Moreover, Mg atoms were more densely distributed in the α-Al matrix due to the high concentration of solution Mg atoms in the alloy matrix, which caused the depinning dislocations to be repinned by solution Mg atoms again soon after slipping off for a short distance. Therefore, the strain cycle of triangular waves corresponding to the low stress flow $\Delta \varepsilon_{T}^{\min}$ on the flow stress–strain curve of the Al-(4.23~9.41) Mg alloy sheets decreased compared to that of the Al-2.86 Mg alloy sheet. Under the comprehensive function of the two aspects mentioned above, the strain cycle $\varepsilon_{T}$ that the flow stress of the alloy sheet oscillated with the flow strain of the Al-(4.23~9.41) Mg alloy sheets was narrow. In other words, the serration oscillation frequency of the flow stress “climb-drop-climb” repeatedly with the flow strain was high, and it gradually increased with increasing strain. Therefore, the stress–strain curve of the Al-(4.23~9.41) Mg alloy sheets was type C. It was noticeable that there was no zigzag oscillation in the flow stress–strain curve of the Al-9.41 Mg alloy sheet in the strain range from 0.0147 to 0.142. The high concentration and dispersed solution Mg atoms in the α-Al matrix and the solution Mg atom diffusion velocity quickened at the initial stage due to the increase in vacancy concentration caused by tensile deformation at room temperature, which increased the segregation of solution Mg atoms on dislocations. The pinning force of solution Mg atoms on dislocations was significantly increased so that the moving dislocations could not break free from the pinning of numerous solute atoms; that is, the moving dislocations were pinned steadily by the solution Mg atoms, and the moving dislocations slip together with the numerous segregation solute atoms. As a result, the stress of the alloy sheet increased continuously and monotonously with increasing strain in the strain range from 0.0147 to 0.142, which is consistent with a previous observation [9]. The Al-9.41Mg alloy sheet had the highest strength among all of the Al-(2.86~9.41) Mg alloy sheets. The higher flow stress in the strain range from 0.0147 to 0.142 during tensile deformation at room temperature will continuously activate the static dislocations, while dislocation multiplication will continuously generate new dislocations. Moreover, dislocation multiplication was enhanced with increasing flow stress, but both the moving dislocations and dislocation multiplication were pinned in a steady state by the numerous segregation solute atoms. Therefore, the flow stress of the Al-9.41 Mg alloy sheet increased continuously and monotonously with increasing strain in the strain range from 0.0147 to 0.142 during tensile deformation at room temperature, and there was no zigzag oscillation in the flow stress–strain curve of the alloy sheet. The vacancy concentration in the Al-9.41Mg alloy sheet almost achieved an equilibrium concentration when the tensile strain exceeded 0.142. The number of segregation Mg atoms on dislocations and the pinning force of the segregation Mg atoms on moving dislocations both maintained dynamic equilibrium, which resulted in the diffusion velocity of solution Mg atoms remaining unchanged. However, the flow stress continued to increase, and the dislocation further increased in density. As a result, it is difficult for solution Mg atoms to keep steady pinning on the continuedly increased moving dislocations. Therefore, the moving dislocation will break free from the pinning of segregation solution Mg atoms and begin to slip rapidly, which leads to a drop in flow stress, and the flow stress–strain curve of the alloy sheet exhibits the PLC phenomenon or zigzag oscillation. This is similar to the results of previous studies by Fu et al. [23].

Furthermore, the strain cycle of the rectangular wave corresponding to high stress flow $\Delta \varepsilon_{T}^{\max}$ and the stress drop amplitude Δσ of quenched Al-(2.86~9.41) Mg alloy sheets increased with increasing the Mg content at the same strain stages during tensile deformation at room temperature (as shown in Figure 6). This is due to the small lattice distortion of the α-Al matrix and low pinning force on dislocations for the quenched Al-2.86 Mg alloy sheet due to the very dilute concentration of solution Mg atoms. Therefore, the stress drop amplitude Δσ of the PCL effect was small, the moving dislocations easily broke free from the pinning of the small quantity of solute Mg atoms, and the strain cycle of the rectangular waves corresponding to high stress flow $\Delta \varepsilon_{T}^{\max}$ on the tensile curve was narrow. With the increase in the Mg content, the concentration of solution Mg atoms in the α-Al matrix increased, the lattice distortion of the α-Al matrix increased, and the pinning force of segregation Mg atoms on moving dislocations increased obviously, which resulted in the stress drop amplitude Δσ in the PCL effect increasing. Meanwhile, it is more difficult for the moving dislocation to break free from pinning by the segregation solution of Mg atoms, and the strain cycle of the rectangular waves corresponding to the high stress flow $\Delta \varepsilon_{T}^{\max}$ in the tensile curve of Al-Mg alloy sheets was enlarged with increasing the Mg content. With increasing strain, the strain cycle of the rectangular waves corresponding to the high stress flow $\Delta \varepsilon_{T}^{\max}$ and the stress drop amplitude Δσ in the tensile curve of the Al-(2.86~9.41) Mg alloy sheets first increased gradually and then remained unchanged (as shown in Figure 7). When the quenched Al-(2.86~9.41) Mg alloy sheets were stretched at room temperature, the flow strain of the alloy sheets increased due to dislocation slip and dislocation multiplication. Meanwhile, the vacancy concentration in the α-Al matrix increased with increasing strain, which accelerated the diffusion of solution Mg atoms in the α-Al matrix. As a result, the number of segregation solution Mg atoms on the movable dislocation increased gradually, and the pinning force of the segregation solution Mg atoms to the moving dislocations also increased. It is difficult for the moving dislocations to break free from the pinning of the segregation solution Mg atoms. Therefore, the strain cycle of the rectangular waves corresponding to the high stress flow $\Delta \varepsilon_{T}^{\max}$ and the stress drop amplitude Δσ on the tensile curve of the alloy sheet increased gradually with increasing tensile deformation. As the tensile deformation increased to the later stage (ε > 0.065), the diffusion velocity of Mg atoms in the α-Al matrix became stable because the vacancy concentration in the α-Al matrix reached saturation, and the pinning force of the segregation solution Mg atoms on the moving dislocations remained unchanged with increasing strain. Therefore, both the strain cycle of the rectangular waves corresponding to high stress flow $\Delta \varepsilon_{T}^{\max}$ and the stress drop amplitude Δσ in the tensile curve of alloy sheets remained unchanged at the late tensile stage.

Previous studies have shown that both the tensile rate [24] and tensile temperature [22] affect the critical strain of the PLC effect in Al-Mg alloys; furthermore, the concentration of solution Mg atoms also has a significant impact on the critical strain of the PLC effect. The results of Zhao et al. [17] showed that the critical strain corresponding to the occurrence of the PLC shearing band $\varepsilon_{c}$ of quenched Al-(0.5~5.0) Mg alloy sheets increased significantly with increasing the Mg content. This is consistent with the phenomenon that the critical deformation of quenched Al-(2.86~9.41) Mg alloy sheets increased with increasing the Mg content when the PLC effect occurred on the tensile curves. However, the yield strength of quenched Al-(2.86~9.41) Mg alloy sheets increased with increasing the Mg content due to the solution strengthening of Mg atoms, and the critical strain corresponding to the yield stress $\varepsilon_{\sigma }$ and the critical strain corresponding to the occurrence of the PLC shearing band $\varepsilon_{c}$ also increased with increasing the Mg content. To objectively and accurately analyze the influence of the number of solution Mg atoms on the critical strain corresponding to the occurrence of the PLC shearing band $\varepsilon_{c}$ in alloy sheets, in the present work, the flow strain difference $\Delta \varepsilon_{c-\sigma }$ = ($\varepsilon_{c}-{\;\varepsilon }_{\sigma }$) was first proposed to describe the characteristics of the PLC effect and to reveal the law of action of the solution Mg atom concentration on the tensile PLC effect of aluminum alloy sheets. $\Delta \varepsilon_{c-\sigma }$ is the difference between the critical strain $\varepsilon_{c}$ of the occurrence of the PLC effect and the critical strain $\varepsilon_{\sigma }$ corresponding to yield. The results showed that both the critical strain corresponding to the occurrence of the PLC shearing band $\varepsilon_{c}$ and the critical strain corresponding to the yield stress $\varepsilon_{\sigma }$ increased with increasing the Mg content. However, the flow strain difference $\Delta \varepsilon_{c-\sigma }$ decreased gradually with increasing the Mg content (as shown in Figure 8). This indicates that the PLC effect of the Al-(2.86~9.41) Mg alloy sheets appears after yielding, and the strain or tensile deformation from the yield point to the occurrence of the PLC effect of the Al-(2.86~9.41) Mg alloy sheets increases with increasing the Mg content. With increasing the Mg content, the concentration of solution Mg atoms in the α-Al matrix and the lattice distortion of the α-Al matrix all increased, which resulted in an increase in both the critical strain corresponding to the yield stress $\varepsilon_{\sigma }$ and the critical strain corresponding to the occurrence of the PLC effect $\varepsilon_{c}$ of the alloy sheet. Only a certain number of solution Mg atoms segregated on the moving dislocations will result in “Pinned-Unpinned-Pinned” moving dislocations in Al-(2.86~9.41) Mg alloy sheets. When the alloy sheets had a low Mg content, the number of solution Mg atoms in the alloy matrix was lower, and the number of segregation solution Mg atoms on the moving dislocations was much lower. The moving dislocations will free slip for a wide strain range until the number and pinning force of segregation solution Mg atoms reach a critical value in the Al-Mg alloy sheet with low Mg content, and the zigzag oscillations on the tensile curve of the alloy sheet will not appear soon after the yield point.

Dislocation multiplication and dislocation density continuously increased as the tensile deformation of the alloy sheet increased, while the number of solution Mg atoms that underwent segregation on the moving dislocations increased gradually. When the number and pinning force of segregation solution Mg atoms reached a critical value, there was an effective pinning effect on the moving dislocations. Then, the phenomenon of stress serrated oscillations with increasing strain appears on the tensile curve of the alloy sheet. Therefore, the concentration of solution Mg atoms required for effective pinning on the movable dislocations $\Delta \varepsilon_{c-\sigma }$ was larger in alloy sheets with lower Mg content than in alloy sheets with higher Mg content. With the increase in the Mg content, the number of solution Mg atoms both in the alloy matrix and segregated on the moving dislocations increased gradually, which resulted in the pinning force of solution Mg atoms on the movable dislocations gradually increasing, and the alloy sheet was subjected to a small tensile deformation before the occurrence of the PLC effect on the tensile curve. This is the fundamental reason why the flow strain range from the yield point to the occurrence of the PLC phenomenon of the alloy sheet decreased gradually with increasing the Mg content. Therefore, the flow strain difference $\Delta \varepsilon_{c-\sigma }$ of the Al-2.86 Mg alloy sheet is the largest, and that of the Al-9.41 Mg alloy sheet is the smallest.

## 5. Conclusions

(1)With an increase in the Mg content, the types of room temperature tensile stress–strain curves of quenched Al-(2.86–9.41) Mg alloy sheets were different: the stress–strain curve of the Al-2.86 Mg alloy sheet was type B + C, while those of the Al-(4.23~9.41) Mg alloy sheets were type C. The yield stress of the alloy sheet increased gradually, and the critical strain corresponding to yield stress $\varepsilon_{\sigma }$ and the critical strain corresponding to the occurrence of the PLC shearing band $\varepsilon_{c}$ both increased. However, the difference in the flow strain value $\Delta \varepsilon_{c-\sigma }$ between $\varepsilon_{c}$ and $\varepsilon_{\sigma }$ of the alloy sheets decreased gradually.(2)When the quenched Al-(2.86~9.41) Mg alloy sheets were stretched at room temperature, the strain cycle of the rectangular waves corresponding to the high stress flow $\Delta \varepsilon_{T}^{\max}$ and the stress drop amplitude Δσ on the zigzag stress–strain curve of alloy sheets both increased with increasing the Mg content.(3)When the quenched Al-(2.86~9.41) Mg alloy sheets were stretched at room temperature, the strain cycle of the rectangular waves corresponding to the high stress flow $\Delta \varepsilon_{T}^{\max}$ and the stress drop amplitude Δσ on the zigzag stress–strain curve of alloy sheets both increased with increasing the tensile deformation.

## Figures and Tables

**Figure 1 materials-15-04965-f001:**
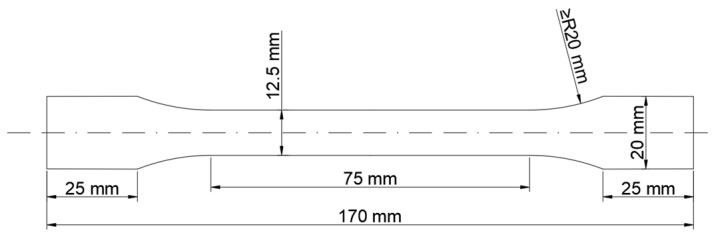
Dimensions of the 1.5-mm-thick tensile specimens.

**Figure 2 materials-15-04965-f002:**
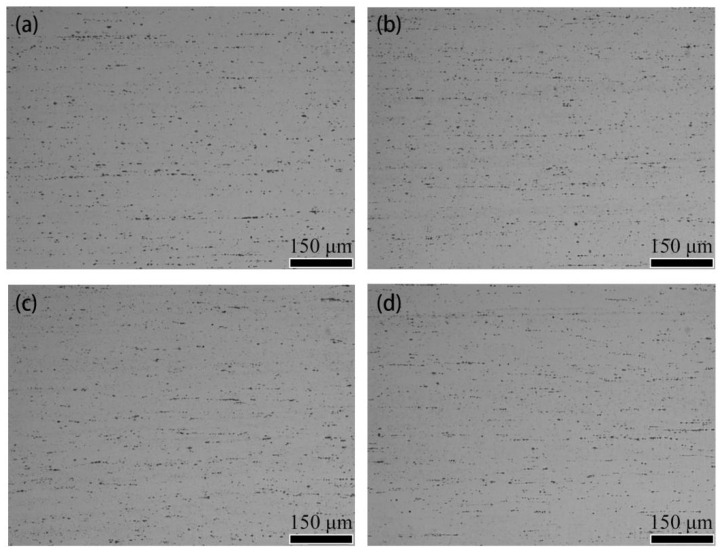
Optical microstructure of quenched Al-(2.86~9.41) Mg alloy sheet with different Mg contents (undecked). (**a**) x = 2.86% Mg; (**b**) x = 4.23% Mg; (**c**) x = 6.51% Mg; (**d**) x = 9.41% Mg.

**Figure 3 materials-15-04965-f003:**
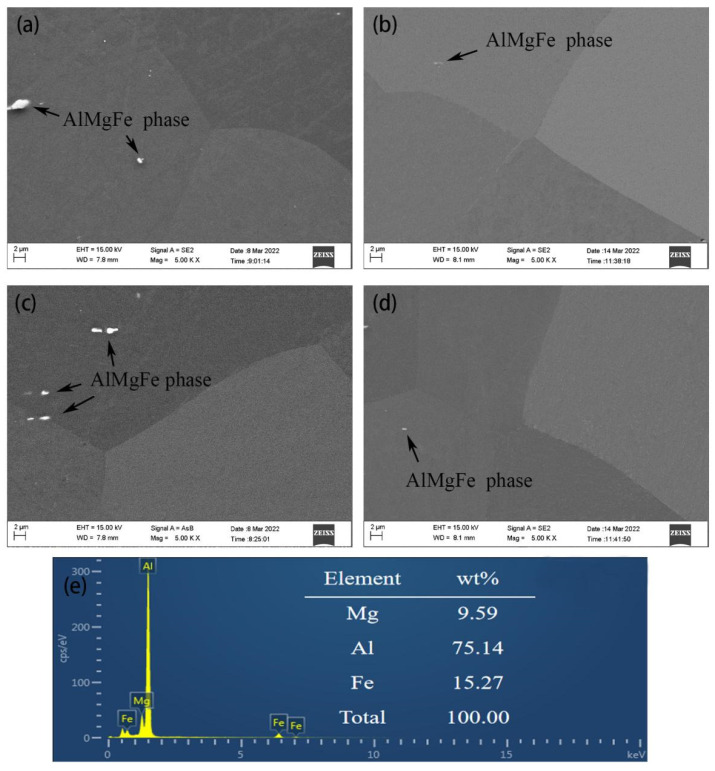
SEM images and EDS results of the Al-(2.86~9.41) Mg alloy sheet (**a**) x = 2.86% Mg; (**b**) x = 4.23% Mg; (**c**) x = 6.51% Mg; (**d**) x = 9.41% Mg. (**e**) the result of EDS analysis of Al Mg Fe phase in the Al-4.23 Mg alloy sheet.

**Figure 4 materials-15-04965-f004:**
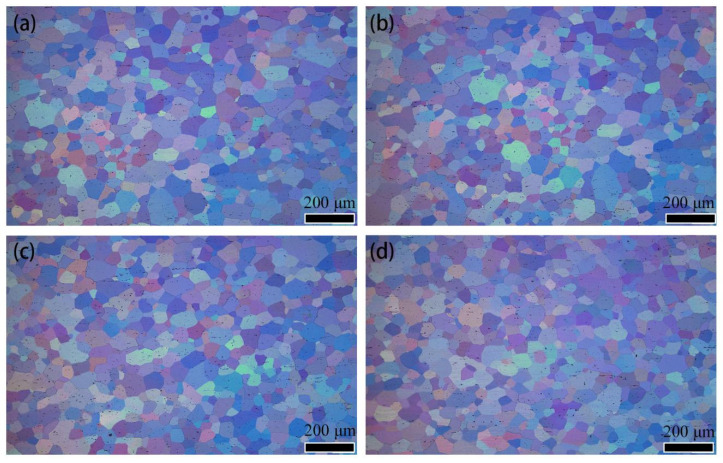
Grain structure of quenched Al-(2.86~9.41) Mg alloy sheet (**a**) x = 2.86%; (**b**) x = 4.23%; (**c**) x = 6.51%; (**d**) x = 9.41%.

**Figure 5 materials-15-04965-f005:**
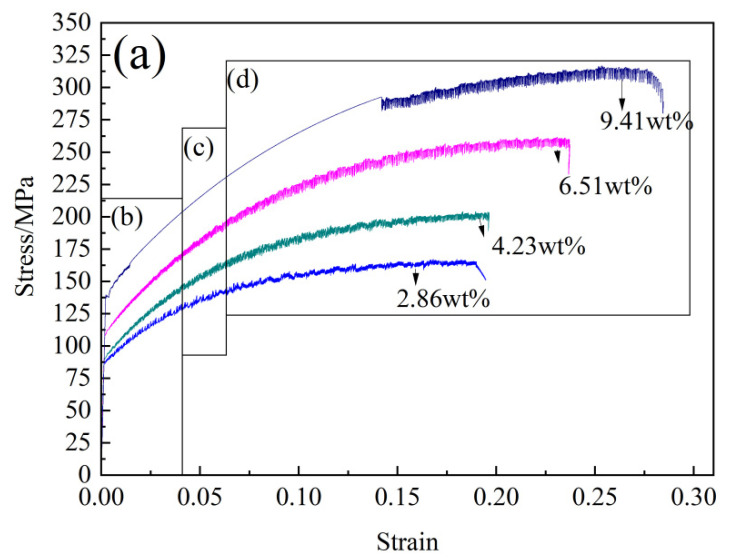
Stress–strain curve of quenched Al-(2.86~9.41) Mg alloy sheet at room temperature at 1 × 10^−3^ s^−1^ strain rate (**a**) Stress–strain curve; (**b**–**d**) are subgraphs of (**a**). (**e**) Enlarged view of initial plastic deformation; (**f**) Enlarged view of initial plastic deformation. (**g**) Enlarged view of the initial plastic deformation; (**h**) Change curve of $\Delta \varepsilon_{T}^{\max}$ with different Mg contents during the plastic deformation stage.

**Figure 6 materials-15-04965-f006:**
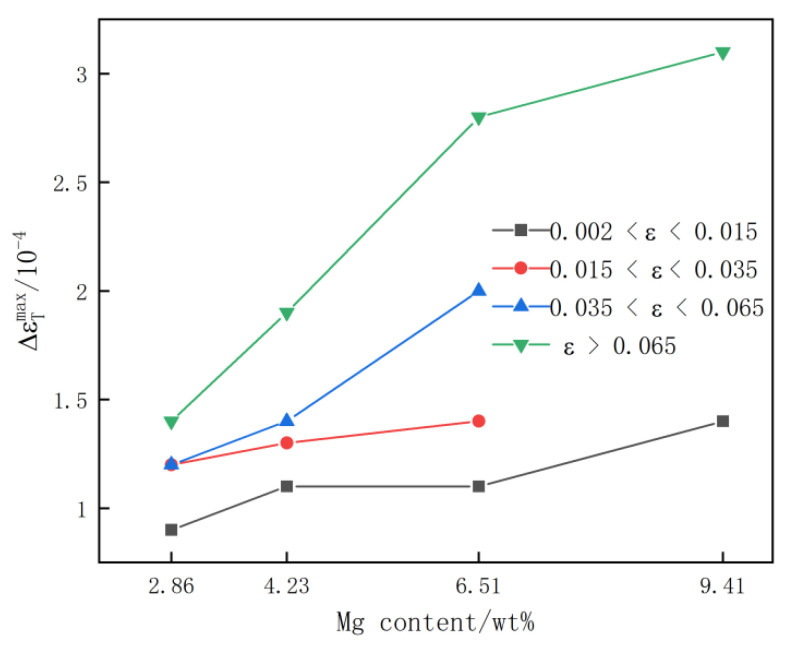
The change in the strain cycle of rectangular waves corresponding to high stress flow of PLC curves of quenched Al-(2.86~9.41) Mg alloy sheet deformation with increasing Mg content.

**Figure 7 materials-15-04965-f007:**
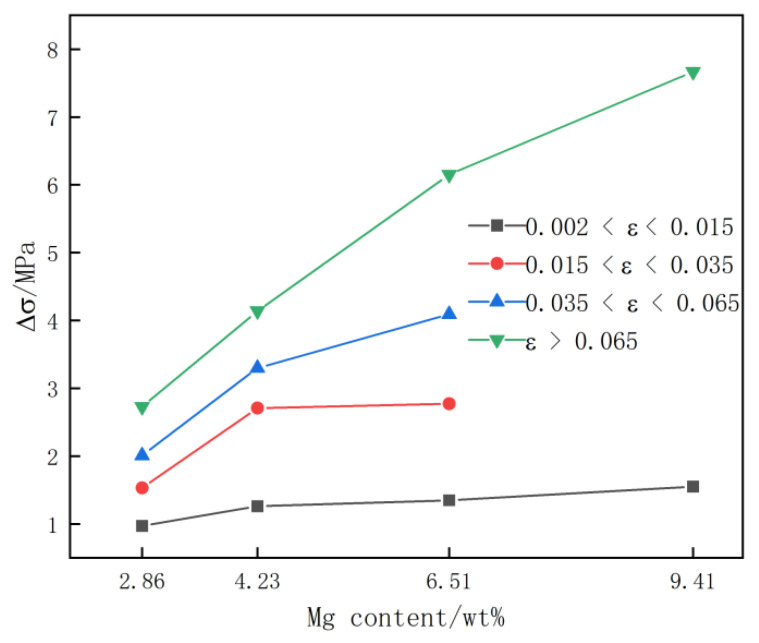
Stress amplitude Δσ curves of the quenched Al-(2.86~9.41) Mg alloy sheet with increasing Mg content.

**Figure 8 materials-15-04965-f008:**
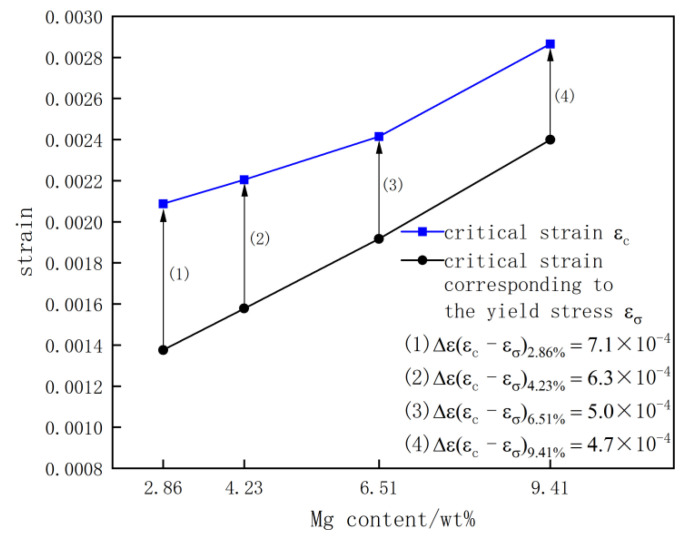
Curves of $\varepsilon_{c}$, $\varepsilon_{\sigma }$ and Δε of the quenched Al-(2.86~9.41) Mg alloy sheet with increasing Mg content.

**Table 1 materials-15-04965-t001:** Chemical compositions of the aluminum alloys used in this research (wt%).

Alloys	Mg	Fe	Si	Zn	Other	Al
1	2.86	0.08	0.02	0.02	<0.01	Bal
2	4.23	0.09	0.02	0.01	<0.01	Bal
3	6.51	0.08	0.01	0.01	<0.01	Bal
4	9.41	0.08	0.02	0.02	<0.01	Bal

## Data Availability

The data presented in this study are available on request from the corresponding author.

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
