# Peer review of "Portevin-Le Chatelier Characterization of Quenched Al-Mg Alloy Sheet with Different Mg Concentrations"

_materials, 2022, doi:10.3390/ma15144965_

Round 1

Reviewer 1 Report

Reviewer comments on manuscript: “ Portevin-Le Chatelier characterization of quenched Al-Mg alloy sheet with different Mg concentrations”

The reviewed manuscript may be published after minor revision.

The reviewed manuscript focused on the Portevin-Le Chatelier characteristic parameters and dynamic strain aging of four designed aluminium alloys of different Mg concentration: 2.86, 4.23, 6.51, 9.41 wt%. Manuscript is very interesting and consist a large “state of study” and “discussion” prats.

Reviewer has noticed some issues in the text which should be cleared:

Lines 112-115; It is not specified what was the casting temperature? Was it the same for each alloy?

Lines 326, 327: an “amplitude” word repeats

Figures description are not formatted uniformly in the whole document – please correct.

There should be distance after figure 2, 6 and 7 description.

It could be a good idea to use numbers to identify the alloy, as in table 1. Using percentage description, as in figure 5 is imprecise as there is no information about what percent (wt, at.), and percent of what (element, phase,…), it refers to.

There is no DOI information in the references.

Author Response

非常感谢您对稿件的宝贵意见,我已完成对所有建议的回答,并对稿件进行了修改。请参阅附件。

Reviewer 2 Report

1) The rewiever suggests to improve the last part of the introduction when the autors describe the present study.

2) The rewiever suggests to improve the conclusion, with a brief summary of the activity done.

Author Response

Thank you very much for your comments on the manuscript, I have replied to each of them and completed the changes in the manuscript. Please see the attachment.

Reviewer 3 Report

The paper: Portevin-Le Chatelier characterization of quenched Al-Mg alloy sheet with different Mg concentrations present some interesting results in the field of Al-based alloys and applications, especially based on them mechanical solicitation.  

L14: the PLC characteristic parameters and DSA mechanism - explain PLC and DSA 

L20: Al-(2.86~9.41)Mg what are the Mg percentages wt or at% ?

L39: [1, 2, 3] - use [1-3] 

at ref. 8 the main author is : Chatterjee , mention him in text at line 51-52 

use few (2-3) newer references in the Introduction section to confirm the novelty of the research 

L115: use a dot . after Table 1. 

In Table 1 : how as the chemical composition determined ?, how many determinations , are these values average values ? provide a StDev or an error of the equipment 

L167: how do you confirm the presence of : AlMgFe phase , beside reference 4 , an XPS, AES or XRD experiment  ? 

L181: images and EDS results : what EDS results ? - please highlight them, also mention in section 2 the EDS detector (type, mode you use, number of determinations, detector error etc.) 

Why is in figure 2 so many AlMgFe phase points and in SEM images only few ?, 

L188: ref 18 is : Nogueira, D. C. R.; Hopperstad, O. S.; Engler, O. ..... why the authors mention Codes in text ? 

Paragraph from the lines: 256-283 needs a reference or additional experiments to prove the atoms slip or move 

L366: Micron ? 

Author Response

(The authors gave the same response as above.)

Reviewer 4 Report

In this work, the influence of the solute Mg atom concentration on the PLC effect of the Al-Mg alloy sheet was investigated to reveal the mechanism of dynamic strain aging effect in a high-magnesium aluminum alloy sheet. The manuscript was well organized and extensively discussed. However, some revisions must be made:

1.      How to accurately evaluate the solute atom concentration in the sample? It is a lack of necessary experimental data.

2.      More Refs in recent years need to be added instead of old ones such as those in the years before 2010.

3.      The first para. In the introduction is too long for readers to follow. Please revise it.

4.      The following refs are recommended to be cited. Materials Science and Engineering: A 808 (2021) 140864; Materials Characterization 171 (2021) 110794.

Author Response

(The authors gave the same response as above.)

Round 2

Reviewer 3 Report

Agree with publication in the current form